# Continuous Depth Recurrent Neural Differential Equations

## Abstract

Recurrent neural networks (RNNs) have brought a lot of advancements in sequence labeling tasks and sequence data. However, their effectiveness is limited when the observations in the sequence are irregularly sampled, where the observations arrive at irregular time intervals. To address this, continuous time variants of the RNNs were introduced based on neural ordinary differential equations (NODE). They learn a better representation of the data using the continuous transformation of hidden states over time, taking into account the time interval between the observations. However, they are still limited in their capability as they use the discrete transformations and a fixed discrete number of layers (depth) over an input in the sequence to produce the output observation. We intend to address this limitation by proposing RNNs based on differential equations which model continuous transformations over both depth and time to predict an output for a given input in the sequence. Specifically, we propose continuous depth recurrent neural differential equations (CDR-NDE) which generalizes RNN models by continuously evolving the hidden states in both the temporal and depth dimensions. CDR-NDE considers two separate differential equations over each of these dimensions and models the evolution in the temporal and depth directions alternatively. We also propose the CDR-NDE-heat model based on partial differential equations which treats the computation of hidden states as solving a heat equation over time. We demonstrate the effectiveness of the proposed models by comparing against the state-of-the-art RNN models on real world sequence labeling problems and data.

## 1 Introduction

Deep learning models such as ResNets (He et al., 2016) have brought a lot of advances in many real world computer vision applications (Ren et al., 2017; He et al., 2020; Wang et al., 2019). They managed to achieve a good generalization performance by addressing the vanishing gradient problem in deep learning using skip connections. Recently, it was shown that the transformation of hidden representations in the ResNet block is similar to the Euler numerical method (Lu et al., 2018; Haber & Ruthotto, 2017) for solving ordinary differential equations (ODE) with constant step size. This observation has led to the inception of new deep learning architectures based on differential equations such as neural ODE (NODE) (Chen et al., 2018). NODE performs continuous transformation of hidden representation by treating Resnet operations as an ODE parameterized by a neural network and solving the ODE using numerical methods such as Euler method and Dopri-5 (Kimura, 2009). NODE automated the model selection (depth estimation), is parameter efficient and is robust towards adversarial attacks than a ResNet with similar architecture (Hanshu et al., 2019).

Recurrent neural networks and its variants such as long short term memory (LSTM) (Hochreiter & Schmidhuber, 1997) and gated recurrent units (GRU) (Cho et al., 2014) were successful and effective in modeling time-series and sequence data. However, RNN models were not effective for irregularly sampled time-series data (Rubanova et al., 2019b), where the observations are measured at irregular intervals of time. ODE-RNN (Rubanova et al., 2019b) modeled hidden state transformations across time using a NODE, where the transformations of hidden representations depended on the time-gap between the arrivals and this led to a better representation of hidden state. This addressed the drawbacks of the RNN models which performs a single transformation of the hidden representation at the observation times irrespective of the time interval. Such continuous recurrent models such as

GRU-ODE (De Brouwer et al., 2019) and ODE-LSTM (Lechner & Hasani, 2020) were proposed to learn better representation of irregular time series data.When applied to the sequence data with a sequence of input-output elements along with their time of occurrences, these models obtain the temporal evolution of hidden states using a neural ODE. At an observation time, this is then combined with the input at that time and a discrete number of transformations is applied using a feed-forward neural network to obtain the final hidden representation. This final hidden representation is then used to produce the desired output. Though these models evolve continuously over time, they use a fixed discrete transformations over depth.

There are several real world sequence labelling problems where the sequences could be of different complexities or the input elements in the sequence could be of different complexities. For instance, consider the problem of social media post classification where different posts arrive at irregular time intervals. The posts could have varying characteristics with some posts containing only text while some contains both text and image. It would be beneficial to have a recurrent neural network model which would consider the complexities of the input in a sequence by having a varying number of transformations for different inputs. In this work, we propose continuous depth recurrent neural differential equation (CDR-NDE) models which generalize the recurrent NODE models to have continuous transformation over depth in addition to the time. Continuous depth allows flexibility in modeling sequence data, with different depths over the elements in the sequence as well as different sequences. Combining this with the continuous time transformation as in recurrent neural ODE allows greater modeling capability for irregularly sampled sequence data.

The proposed continuous depth recurrent neural differential equations (CDR-NDE) model the evolution of the hidden states simultaneously in both the temporal and depth dimensions using differential equations. Continuous transformation of hidden states is modeled as a differential equation with two independent variables, one in the temporal and the other in the depth direction. We also aim to model the evolution of the hidden states using a partial differential equation (PDE) based on the 1D-heat equation, leading to the CDR-NDE-heat model. Heat equation is a second order partial differential equation, which models the flow of heat across the rod over time. The proposed CDR-NDE-heat model considers the transformation of hidden states across depth and time using a non-homogeneous heat equation. An advantage is that it is capable of considering the information from the future along with the past in sequence labeling tasks. We exploit the structure in the CDR-NDE-heat model and PDE solvers to develop an efficient way to obtain the hidden states where all the hidden states at a particular depth can be computed simultaneously. We evaluate the performance of our proposed models on real-world datasets such as person activity recognition (Asuncion & Newman, 2007) and Walker2d kinematic simulation data (Lechner & Hasani, 2020). Through experiments, we show that the proposed continuous depth recurrent neural differential equation models outperformed the state-of-the-art recurrent neural networks in all these tasks.

## 2 RELATED WORK

RNN models such as LSTM (Hochreiter & Schmidhuber, 1997) and GRU (Cho et al., 2014) are the primary choice to fit high-dimensional time-series and sequence data. For irregular time-series data, traditional LSTM and GRU models are less effective as they do not consider the varying inter-arrival times. To address the problem of fitting irregular time-series data, the standard approach is the augmented-LSTM which augments the elapsed time with input data. In GRU-D (Che et al., 2018) and RNN-Decay (Rubanova et al., 2019b), the computed hidden state is the hidden state multiplied by a decay term proportional to the elapsed time. In other variants such as CT-GRU (Mozer et al., 2017),CT-RNN (Funahashi & Nakamura, 1993), ODE-RNN (Rubanova et al., 2019b), GRU-ODE (De Brouwer et al., 2019), ODE-LSTM (Lechner & Hasani, 2020) and Jump-CNF (Chen et al., 2020), the hidden state is computed as a continuous transformation of intermediate hidden states. CT-LSTM (Mei & Eisner, 2017a) combines both LSTM and continuous time neural Hawkes process to model continuous transformation of hidden states. Two alternative states are computed at each time-step and the final state is an interpolated value of these hidden states, where the interpolation depends on the elapsed time. Phased-LSTM (Neil et al., 2016) models irregularly sampled data using an additional time gate. The updates to the cell state and hidden state only happen when the time gate is open. This time gate allows for the updates to happen at irregular intervals. Phased LSTM reduces the memory decay as the updates only happen in a small time when the time gate is open. ODE-RNN (Rubanova et al., 2019b) used neural ordinary differential equations over time to model the evolution of the hidden

states. The next hidden state is obtained as a solution to a NODE, and depends on the time interval between two consecutive observations. GRU-ODE (De Brouwer et al., 2019) derived a NODE over time and hidden states using the GRU operations and consequently could avoid the vanishing gradient problem in ODE-RNN. Similarly, ODE-LSTM (Lechner & Hasani, 2020) addressed the vanishing gradient problem in ODE-RNN by considering the LSTM cell and memory cell while the output state is modeled using a neural ODE to account for irregular observations. However, all these models only considered continuous evolution of hidden states over the temporal dimension. In our work, we aim to develop models which consider continuous evolution of the hidden states over depth as well as temporal dimensions.

Recently, there are some works which used deep neural networks to solve the partial differential equations (PDE) (also known as neural PDEs or physics informed neural networks) (Zubov et al., 2021; Brandstetter et al., 2021; Hu et al., 2020). (Hu et al., 2020) showed that LSTM based RNNs can efficiently find the solutions to multidimensional PDEs without knowing the specific form of PDE. On the other hand, very few works used PDEs to model DNN architectures for solving problems from any domain. (Ruthotto & Haber, 2018) used PDE to design Resnet architectures and convolutional neural networks (CNNs) such as Parabolic CNN and hyperbolic CNN by changing the ODE update dynamics to different PDE update dynamics. For instance, hyperbolic CNN can be obtained with second order dynamics. They showed that even the PDE CNNs with modest architectures achieve similar performance to the larger networks with considerably large numbers of parameters. Unlike the prior works combining neural networks and PDEs, our aim is to solve sequence labeling problems by developing flexible RNN based architectures considering the PDE based models and solutions.

## 3 BACKGROUND

### 3.1 PROBLEM DEFINITION

We consider the sequence labeling problem with a sequence length of $K$, and denote the input-output pairs in the sequence as $\{\mathbf{x}_t, \mathbf{y}_t\}_{t=1}^{K}$ and the elements in the sequence are irregularly sampled at observation times $\mathbf{t} \in \mathbb{R}^{+K}$. We assume the input element in the sequence to be $D$ dimensional, $\mathbf{x}_t \in \mathcal{R}^D$ and the corresponding output $\mathbf{y}_t$ depends on the problem, discrete if it is classification or continuous if it is regression. The aim is to learn a model $f(\cdot, \theta)$ which could predict the output $\mathbf{y}_t$ considering the input $\mathbf{x}_t$, and dependence on other elements in the sequence.

### 3.2 GATED RECURRENT UNIT

Recurrent neural networks (RNNs) are well suited to model the sequence data. They make use of the recurrent connections to remember the information until the previous time step, and combine it with the current input to predict the output. Standard RNNs suffer from the vanishing gradient problem due to which it forgets long term dependencies among the sequence elements. This was overcome with the help of long short term memory (LSTM) (Hochreiter & Schmidhuber, 1997) and gated recurrent units (GRUs) (Cho et al., 2014). In our work we consider the basic RNN block to be a GRU. In GRU, computation of hidden state and output at any time step $t$ involves the following transformations,

$$\mathbf{r}_t = \sigma(W_r \mathbf{x}_t + U_r \mathbf{h}_{t-1} + \mathbf{b}_r), \quad \mathbf{z}_t = \sigma(W_z \mathbf{x}_t + U_z \mathbf{h}_{t-1} + \mathbf{b}_z)$$
$$\mathbf{g}_t = \tanh(W_h \mathbf{x}_t + U_h(\mathbf{r}_t \odot \mathbf{h}_{t-1}) + \mathbf{b}_h)$$
(1)

Where $\mathbf{r}_t, \mathbf{z}_t, \mathbf{g}_t$ are the reset gate, update gate and update vector respectively for the GRU. The hidden state $\mathbf{h}_t$ in GRU is given by,

$$\mathbf{h}_t = \mathbf{z}_t \odot \mathbf{h}_{t-1} + (1 - \mathbf{z}_t) \odot \mathbf{g}_t$$
(2)

As we can see, GRUs and RNNs in general do not consider the exact times or time interval between the observations. The same operations are applied irrespective of the time gap between observations. This can limit the capability of these models for irregularly sampled time series data.

GRUs can be extended to consider the irregularity in the time series data by developing a continuous GRU variant. A continuous GRU, GRU-ODE (De Brouwer et al., 2019), can be obtained by adding and subtracting the hidden state on both sides of equation 2. The computation of hidden states then

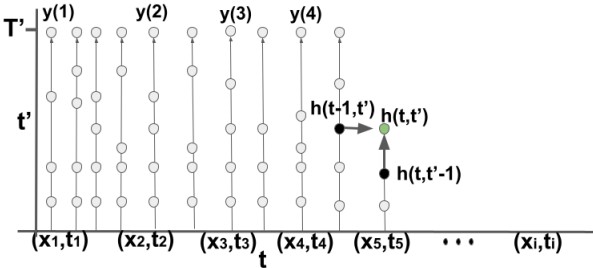

Figure 1: Shows the computation of hidd en states using the CDR-NDE model. As we can see for different observations, the number of intermediate hidden states in the evolution along the depth is different. The index $(t-1, t')$ points to the hidden state of immediate left vertical evolution. The index $(t, t'-1)$ points the hidden state just below the current vertical evolution.

becomes equivalent to solving an ODE given in equation 15.

$$\Delta \mathbf{h}_t = \mathbf{h}_t - \mathbf{h}_{t-1} = \mathbf{z}_t \odot \mathbf{h}_{t-1} + (1 - \mathbf{z}_t) \odot \mathbf{g}_t - \mathbf{h}_{t-1} \implies \frac{d\mathbf{h}_t}{dt} = (1 - \mathbf{z}_t) \odot (\mathbf{g}_t - \mathbf{h}_t) \quad (3)$$

### 3.3 RECURRENT NEURAL ORDINARY DIFFERENTIAL EQUATIONS

In ODE-RNN (Rubanova et al., 2019a), ODE-LSTM (Lechner & Hasani, 2020), and ODE-GRU (De Brouwer et al., 2019), the hidden state $\mathbf{h}_t$ holds the summary of past observations and evolves continuously between the observations considering the time interval. When there is a new observation, the hidden state $\mathbf{h}_t$ changes abruptly (Chen et al., 2020) to consider it. Given initial state $\mathbf{h}_0$, let the function $f_h()$ models the continuous transformation of hidden state and function $g_h()$ model instantaneous change in the hidden state at the new observation. The prediction $y'_t$ is computed by using a function $o_h()$, which is then used to compute the loss (cross entropy loss for classification). Both the functions $g_h()$ and $o_h()$ are typically standard feed-forward neural networks with a discrete number of layers. In the case of GRU-ODE, the function $f_h()$ takes the form given in the right hand side of equation 15. In general, the recurrent NODE models can be represented using the following system.

$$\frac{d\mathbf{h}_t}{dt} = f_h(h_t), \quad \lim_{\epsilon \to 0} \mathbf{h}_{t+\epsilon} = g_h(\mathbf{h}_t, \mathbf{x}_t), \quad y'_t = o_h(\mathbf{h}_{t+\epsilon}) \quad (4)$$

## 4 CONTINUOUS DEPTH RECURRENT NEURAL DIFFERENTIAL EQUATIONS

We propose continuous depth recurrent neural differential equations (CDR-NDE) aiming to overcome the drawbacks of the recurrent NODEs in modeling the sequence data. As already discussed, recurrent NODE models bring abrupt changes in the hidden state at the observation times using the standard neural network transformation $g_h()$, when it considers the input $\mathbf{x}_t$ at time $t$. We aim to develop RNN models capable of continuous transformations over depth in addition to the temporal dimension. Such models will help in processing inputs with varying complexities, aid in model selection (in choosing the number of layers in a RNN model) and reduce the number of parameters (as every layer shares the same parameters as in NODE). Moreover, we hypothesize that such continuous transformation over depth will also aid in learning better hidden representations as it is not limited by a predefined number of layers as in standard RNNs. We propose two models, CDR-NDE and CDR-NDE-heat model. Both the models generalize the GRU-ODE model by continuously evolving in both the temporal and depth directions. The CDR-NDE-heat model is formulated based on a partial differential equation (1-dimensional heat equation) and enables faster computation of hidden states even when using an adaptive step numerical method like Dopri5.

The hidden states of CDR-NDE evolves in depth direction (denoted by vertical axis $t'$ in Figure 1) while also evolving in the temporal direction (denoted by horizontal axis $t$ in Figure 1). As the hidden state evolves in both directions, we need a tuple $(t, t')$ to uniquely identify any hidden state and we represent the hidden state as $\mathbf{h}_{(t,t')}$. As shown in Figure 1, during the computation of a hidden state $\mathbf{h}_{(t,t')}$, it requires hidden states that are immediately to the left in the temporal dimension $\mathbf{h}_{(t-1,t')}$

and below in the depth direction $\mathbf{h}_{(t,t'-1)}$. The evolution of the hidden state $\mathbf{h}_{(t,t')}$ in the CDR-NDE is governed by the following differential equations.

$$\frac{\partial \mathbf{h}_{(t,t')}}{\partial t} = f_h(\mathbf{h}_{(t,t')}, \mathbf{h}_{(t,t'-1)}), \quad \frac{\partial \mathbf{h}_{(t,t')}}{\partial t'} = g_h(\mathbf{h}_{(t,t')}, \mathbf{h}_{(t-1,t')}) \tag{5}$$

$$\mathbf{y}_i = o_h(\mathbf{h}_{(t_i,T')}), \quad \mathbf{h}_{(t_i,0)} = \mathbf{x}_i \quad \forall i = 1, \ldots, K \tag{6}$$

where $T'$ is the maximum depth. We observe that the changes in the hidden state in the horizontal (time) direction depends on the hidden states at a depth below while changes in the hidden states in the vertical (depth) direction depends on the hidden states at the previous time. The derivation and exact expression used to define the functions $f_h()$ and $g_h()$ are obtained as follows. We consider the evolution in the horizontal direction to follow the GRU-ODE model but with an added skip-connection in the vertical direction. Though $f_h()$ can be any function as in ODE-RNN, we followed GRU-ODE to avoid vanishing gradient problems in the temporal direction (De Brouwer et al., 2019). In a discrete setup, the expression used to compute the hidden state $\mathbf{h}_{(t,t')}$ after adding the skip connection in the vertical direction can be written as

$$\mathbf{h}_{(t,t')} = \mathbf{z}_{(t,t')} \odot \mathbf{h}_{(t-1,t')} + (1 - \mathbf{z}_{(t,t')}) \odot \mathbf{g}_{(t,t')} + \mathbf{h}_{(t,t'-1)} \tag{7}$$

By subtracting $\mathbf{h}_{(t-1,t')}$ on both sides, we can obtain the difference equation as

$$\mathbf{h}_{(t,t')} - \mathbf{h}_{(t-1,t')} = \mathbf{z}_{(t,t')} \odot \mathbf{h}_{(t-1,t')} + (1 - \mathbf{z}_{(t,t')}) \odot \mathbf{g}_{(t,t')} + \mathbf{h}_{(t,t'-1)} - \mathbf{h}_{(t-1,t')} \tag{8}$$

Consequently, the differential equation governing the flow in the temporal (horizontal) direction is

$$\frac{\partial \mathbf{h}_{(t,t')}}{\partial t} = \mathbf{z}_{(t,t')} \odot \mathbf{h}_{(t,t')} + (1 - \mathbf{z}_{(t,t')}) \odot \mathbf{g}_{(t,t')} + \mathbf{h}_{(t,t'-1)} - \mathbf{h}_{(t,t')} \tag{9}$$

where

$$\mathbf{z}_{(t,t')} = \sigma(W_z \mathbf{h}_{(t,t'-1)} + U_z \mathbf{h}_{(t,t')} + \mathbf{b}_z), \quad \mathbf{g}_{(t,t')} = \tanh(W_h \mathbf{h}_{(t,t'-1)} + U_h(\mathbf{r}_{(t,t')} \odot \mathbf{h}_{t,t'}) + \mathbf{b}_h)$$
$$\mathbf{r}_{(t,t')} = \sigma(W_r \mathbf{h}_{(t,t'-1)} + U_r \mathbf{h}_{(t,t')} + \mathbf{b}_r)$$

To derive the differential equation in the depth (vertical) direction $t'$, Equation 7 can be written as a difference equation by carrying the term $\mathbf{h}_{(t,t'-1)}$ to the left hand side.

$$\mathbf{h}_{(t,t')} - \mathbf{h}_{(t,t'-1)} = \mathbf{z}_{(t,t')} \odot \mathbf{h}_{(t-1,t')} + (1 - \mathbf{z}_{(t,t')}) \odot \mathbf{g}_{(t,t')} \tag{10}$$

Consequently, the differential equation governing the flow in the depth (vertical) direction is defined below and we can observe that it depends on the hidden states at the previous time.

$$\frac{\partial \mathbf{h}_{(t,t')}}{\partial t'} = \mathbf{z}'_{(t,t')} \odot \mathbf{h}_{(t-1,t')} + (1 - \mathbf{z}'_{(t,t')}) \odot \mathbf{g}'_{(t,t')} \tag{11}$$

where

$$\mathbf{z}'_{(t,t')} = \sigma(W_z \mathbf{h}_{(t,t')} + U_z \mathbf{h}_{(t-1,t')} + \mathbf{b}_z), \quad \mathbf{g}'_{(t,t')} = \tanh(W_h \mathbf{h}_{(t,t')} + U_h(\mathbf{r}'_{(t,t')} \odot \mathbf{h}_{(t-1,t')}) + \mathbf{b}_h)$$
$$\mathbf{r}'_{(t,t')} = \sigma(W_r \mathbf{h}_{(t,t')} + U_r \mathbf{h}_{(t-1,t')} + \mathbf{b}_r)$$

We solve the differential equations equation 9 and equation 11 in two stages. In the first stage, CDR-NDE is solved in the horizontal direction until time $t_K$ for $t' = 0$ following equation 9 and can be solved using differential equation solvers such as Euler method or Dopri5. In the second stage, for every time step evaluated on the $t$-axis during the first stage, hidden states are allowed to evolve in the vertical direction, i.e. along the $t'$-axis. Evolution in vertical direction is done until time $t' = T'$ and can be solved using solvers such as Euler or Dopri5. We can observe that during this evolution, CDR-NDE model considers $\mathbf{h}_{(t-1,t')}$ in computing $\mathbf{h}_{(t,t')}$ for any time t and depth t' taking into account the dependencies in the sequence. Hence, computation of the hidden state $\mathbf{h}_{(t,t')}$ needs access to the hidden state $\mathbf{h}_{(t-1,t')}$ and this requires performing an additional interpolation on the hidden states evaluated at time $t - 1$ in the case of adaptive solvers.

## 4.1 CDR-NDE BASED ON HEAT EQUATION

We propose another CDR-NDE model inspired by the partial differential equations and in particular the 1D-heat diffusion equation (Cannon, 1984). Heat equation represents the evolution of heat over

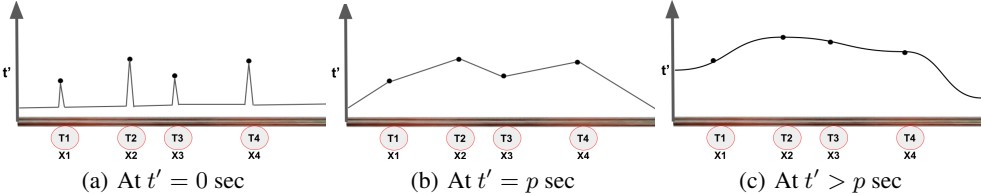

(a) At $t' = 0$ sec      (b) At $t' = p$ sec      (c) At $t' > p$ sec

Figure 2: Evolution of temperature across time.(a) shows the initial state(temperature) of the rod, heat is applied externally at 4 different points. Over time, heat diffuses from hot region to cold region, (b) shows the state of the rod, after $p$ seconds. (c) Over time, the change in temperature comes to an equilibrium state, no change of temperature over time.

time in a rod. Consider a rod of length $L$ which is at room temperature. Along the length of the rod, at different points, heat is applied externally into the rod. The temperatures applied at different points can be different. Figure 2(a) provides a visualization of the initial state(at $t' = 0$), where the rod is at room temperature. At four different points, heat is applied externally with different temperature values. Heat starts to flow from hotter regions to colder regions and the points which are initially at room temperature become hotter. In a finite amount of time, it reaches a state where the change in temperature at any point in time is zero, which is called an equilibrium state. Figure2(b) visualizes the intermediate state of temperatures across the rod after $p$ seconds. Figure2(c) visualizes the equilibrium state where the change in temperature across the rod is smooth.

We can observe that the evolution of temperature in a rod can be seen as equivalent to the evolution of hidden states and applying temperature to the road can be considered equivalent to providing input data in the sequence. The hidden states associated with input data smoothly change over the depth dimension $t'$ reaching an equilibrium state, and finally leading to the output elements in the sequence. The heat equation motivates us to construct a model capable of capturing interactions among different elements in the sequence, providing smooth hidden transformations with variable depth. We hypothesize that such models will be able to learn better representations depending on the input and improve the generalization performance.

The process of heat diffusion can be represented by a 1D heat equation (Cannon, 1984) which is a homogeneous second order partial differential equation, $\frac{\partial u(t',l)}{\partial t'} = C * \frac{\partial^2 u(t',l)}{\partial l^2}$, where $u(t',l)$ is the temperature at point $l$ on rod at time $t'$ and $C$ is a constant(diffusivity). The proposed CDR-NDE-heat model is based on the non-homogeneous heat equation, a variant of the homogeneous 1D heat equation. The temperature applied at a location $l_i$ in the rod is equivalent to the datapoint $\mathbf{x}_i$ at time $t_i$. As the temperature injected at a point affects the temperature around the rod neighborhood, the hidden states are affected by the observed data points in the neighborhood. The hidden state then evolves over the depth variable $t'$ and reaches a steady state following the heat diffusion model. The second order derivative with respect to location $l$ in the heat equation (equivalently over time $t$ for sequence labeling problems) allows one to consider the effect of neighbors around a point. For sequence modeling, this allows one to learn a better representation by considering past and future hidden states across time $t$, similar to bi-directional RNNs considering past and future information.

The proposed model considers a non-homogeneous heat equation model which allows a better representation of hidden state during evolution by considering additional information on the interaction between the hidden states. In our case, we choose GRUcell which holds a summary of the past observations to capture the interaction. The differential equation governing the evolution of the proposed CDR-NDE-heat model is defined as follows,

$$\frac{\partial \mathbf{h}_{(t,t')}}{\partial t'} - \frac{\partial^2 \mathbf{h}_{(t,t')}}{\partial t^2} = f(\mathbf{h}_{(t,t'-1)}, \mathbf{h}_{(t-1,t')}) \tag{12}$$

where $f(\mathbf{h}_{(t,t'-1)}, \mathbf{h}_{(t-1,t')})$ is the GRUCell operation, i.e. $f(\mathbf{h}_{(t,t'-1)}, \mathbf{h}_{(t-1,t')}) = \mathbf{z}_{(t,t')} \odot \mathbf{h}_{(t-1,t')} + (1 - \mathbf{z}_{(t,t')}) \odot \mathbf{g}_{(t,t')}$. The evolution of hidden state as shown in Equation12 corresponds to a non-homogeneous heat equation (Trong et al., 2005) with GRUCell capturing the interactions.

The heat equation can be solved numerically using methods like finite-difference method(FDM) (Recktenwald, 2004) and method of lines(MoL) (Schiesser, 2012). We can get a better insights on the

behaviour of the proposed CDR-NDE-heat model by writing the updates using the finite difference method. Using FDM, the hidden state is computed as follows,

$$\frac{\mathbf{h}_{(t,t'+\Delta_{t'})} - \mathbf{h}_{(t,t')}}{\Delta_{(t')}} - \frac{\mathbf{h}_{(t-\Delta_t,t')} - 2\mathbf{h}_{(t,t')} + \mathbf{h}_{(t+\Delta_t,t')}}{\Delta_t^2} = f(\mathbf{h}_{(t,t'-\Delta_{t'})}, \mathbf{h}_{(t-\Delta_t,t')})$$

$$\implies \mathbf{h}_{(t,t'+\Delta_{t'})} = \frac{\Delta_t'}{\Delta_t^2}[\mathbf{h}_{(t-\Delta_t,t')} - 2\mathbf{h}_{(t,t')} + \mathbf{h}_{(t+\Delta_t,t')}] + \Delta_{t'}[f(\mathbf{h}_{(t,t'-\Delta_{t'})}, \mathbf{h}_{(t-\Delta_t,t')})] + \mathbf{h}_{(t,t')}$$

$$(13)$$

FDM divides the space of $(t, t')$ into finite grids as shown in Figure 3. To compute the hidden state at a depth $t' + \Delta_{t'}$, it utilizes the hidden states computed for itself and its immediate neighbors at the previous depth $(\mathbf{h}_{(t-\Delta_t,t')}, \mathbf{h}_{(t,t')}, \mathbf{h}_{(t+\Delta_t,t')})$. This helps to capture dependence among the neighboring inputs during evolution. A drawback of directly using FDM in solving the proposed CDR-NDE-heat model is that it is a slow process. It doesn't exploit the GPU power as the computations are happening in a sequential order.

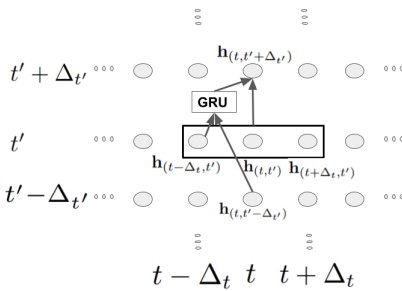

For the proposed model, from the formulation to compute next hidden state in Equation 13 and Figure 3, we can observe that the hidden states at $t' + \Delta_{t'}$ only depends on the hidden states computed below $t' + \Delta_{t'}$. Hence, all the hidden states $t' + \Delta_{t'}$ can be computed simultaneously once we have hidden states at time $t'$. The numerical techniques based on Method of lines

Figure 3: Pictorial representation of computing a hidden state $\mathbf{h}_{(t,t'+\Delta_{t'})}$ as shown in Equation 13. The new hidden state depends on the already computed hidden states in the lower layers.

(MoL) (Schiesser, 2012) is a good choice for such a scenario. MoL method typically discretizes and computes function values in one dimension, and then jointly evolves in the remaining dimension to compute all the function values. In our approach, we first compute hidden states along the $t$-axis and then compute the hidden states along the $t'$-axis by posing as a solution to the system of differential equations. The evolution of the hidden states along the $t'$-axis is defined by the ordinary differential equation (14), which is derived from Equation 13.

$$\frac{\partial \mathbf{h}_{(t,t')}}{\partial t'} = g_\theta(\mathbf{h}_{(t-\Delta_t,t')}, \mathbf{h}_{(t,t')}, \mathbf{h}_{(t+\Delta_t,t')}) = \frac{\mathbf{h}_{(t-\Delta_t,t')} - 2\mathbf{h}_{(t,t')} + \mathbf{h}_{(t+\Delta_t,t')}}{\Delta_t^2} + f(\mathbf{h}_{(t,t')}, \mathbf{h}_{(t-\Delta_t,t')})$$

$$(14)$$

$$\mathbf{h}_{(:,T')} = \text{ODESOLVE}(g_\theta, initial\_state = \mathbf{h}_{(:,0)}, start\_time = 0, end\_time = T')$$

The initial hidden states at $t' = 0$ ($\mathbf{h}_{(:,0)}$) are computed by solving an ODE along the $t$-axis

$$\frac{d\mathbf{h}_{(t,0)}}{dt} = (1 - \mathbf{z}_{(t,0)}) \odot (\mathbf{g}_{(t,0)} - \mathbf{h}_{(t,0)})$$

$$(15)$$

One can select any numerical method for solving the system of ODEs. In the experiments, we evaluate the performance of the CDR-NDE-heat model using both Euler (CDR-NDE-heat(Euler)) and Dopri5 (CDR-NDE-heat(Dopri5)) methods. We can observe that CDR-NDE-heat model considers $\mathbf{h}_{(t+\Delta_t,t')}$ in addition to $\mathbf{h}_{(t-\Delta_t,t')}$ in computing $\mathbf{h}_{(t,t')}$ for any time t and depth t', taking into account more dependencies in the sequence.

After computing the hidden states at depth $T'$, predictions are made using a fully connected neural network, i.e. $\mathbf{y}_i = o_h(\mathbf{h}_{(t_i,T')})$. This is then used to compute loss - cross-entropy for classification and root mean square error for regression problems. The parameters of the CDR-NDE models, i.e. weight parameters of the GRUCell, are learnt by minimizing the loss computed over all the observations in the sequence and over all the sequences. The computed loss is backpropagated using either adjoint method (Chen et al., 2018) or automatic differentiation to update the model parameters.

## 5 EXPERIMENTS

To evaluate the performance of the proposed models, we conduct experiments on irregular time series datasets like person activity recognition (Asuncion & Newman, 2007) and

Table 1: ODE solvers used for different RNODE models. For the CDR-NDE-Heat model using Dopri5, the absolute and relative tolerance values are $1e^{-3}$ and $1e^{-3}$ respectively.

| Model | ODE-Solver | Time-step Ratio |
|---|---|---|
| CT-RNN | 4-th order Runge-Kutta | 1/3 |
| ODE-RNN | 4-th order Runge-Kutta | 1/3 |
| GRU-ODE | Explicit Euler | 1/4 |
| ODE-LSTM | Explicit Euler | 1/4 |
| CDR-NDE | Explicit Euler | 1/2 |
| CDR-NDE-heat(Euler) | Explicit Euler | 1/2 |
| CDR-NDE-heat(Dopri5) | Dopri5 | - |

Table 3: Column 2, shows the test accuracy (mean ± std) of all the models trained on the dataset **Person activity recognition**. Column 3 shows the test data Mean-square error (mean ± std) of all the models trained on the datasets **Walker2d Kinematic**. For both the dataset, every model is trained for 5 times with 5 different seeds.

| Model | Person Activity Recognition Test-Accuracy | Walker2d Kinematic Mean-Square Error(Test-data) |
|---|---|---|
| Augmented-LSTM | $83.78 \pm 0.41$ | $1.09 \pm 0.01$ |
| CT-RNN | $82.32 \pm 0.83$ | $1.25 \pm 0.03$ |
| ODE-RNN | $75.03 \pm 1.87$ | $1.88 \pm 0.05$ |
| ODE-LSTM | $83.77 \pm 0.58$ | $0.91 \pm 0.02$ |
| CT-GRU | $83.93 \pm 0.86$ | $1.22 \pm 0.01$ |
| RNN-Decay | $78.74 \pm 3.65$ | $1.44 \pm 0.01$ |
| Bidirectional-RNN | $82.86 \pm 1.17$ | $1.09 \pm 0.01$ |
| GRU-D | $82.52 \pm 0.86$ | $1.14 \pm 0.01$ |
| Phased-LSTM | $83.34 \pm 0.59$ | $1.10 \pm 0.01$ |
| GRU-ODE | $82.80 \pm 0.61$ | $1.08 \pm 0.01$ |
| CT-LSTM | $83.42 \pm 0.69$ | $1.03 \pm 0.02$ |
| CDR-NDE | $\mathbf{87.54 \pm 0.34}$ | $0.97 \pm 0.04$ |
| CDR-NDE-heat (Euler) | $\mathbf{88.24 \pm 0.31}$ | $0.54 \pm 0.01$ |
| CDR-NDE-heat (Dopri5) | $\mathbf{88.60 \pm 0.26}$ | $\mathbf{0.49 \pm 0.01}$ |

walker2d kinematic simulation (Lechner & Hasani, 2020). We compare our proposed models against RNN models which are designed to tackle the irregularly sampled time-series data. The experimental setup such as the numerical method, hidden state dimension and other hyperparameters is the same as in (Lechner & Hasani, 2020) and is provided in Table 2. Table 1 provides the choice of numerical methods for each model. The proposed models CDR-NDE and CDR-NDE-heat(Euler) used the Euler method with the number of steps as 2. CDR-NDE-heat(Dopri5) used Dopri5 with the absolute and relative tolerance set to $1e^{-3}$. Scheduled learning rate decay is used with decay parameter $\gamma = 0.1$, scheduled at epoch 100. The models are trained on Nvidia Tesla V-100 32GB GPU.

Table 2: Hyperparameter Details

| Parameter | Value |
|---|---|
| Hidden state Dimension | 64 |
| Minibatch size | 256 |
| Optimizer | RMSprop |
| Learning rate | $5e^{-3}$ |
| Training epochs | 200 |

## 5.1 BASELINES

We compared our proposed models against RNN models which are designed to address the problem of fitting irregular time-series data such as GRU-ODE (De Brouwer et al., 2019), CT-GRU (Mozer et al., 2017),CT-RNN (Funahashi & Nakamura, 1993),GRUD (Che et al., 2018),Phased-LSTM (Neil et al., 2016),ODE-LSTM (Lechner & Hasani, 2020), bidirectional-RNN (Schuster & Paliwal, 1997), RNN decay (Rubanova et al., 2019b),Hawk-LSTM (Mei & Eisner, 2017b), Augmented LSTM (Lechner & Hasani, 2020), and ODE-RNN (Rubanova et al., 2019b).

## 5.2 Person activity recognition with irregularly sampled time-series

Dataset contains sensor data from 4 from different sensors(1 for each ankle, 1 chest and 1 belt) attached to 5 different people, performing 5 sequences of activities. The task is to classify the activity based on the sensor data. There are 11 different activities which are reduced to 7 as suggested in (Rubanova et al., 2019b). Dataset is transformed such that each step in the recording contains 7 values (4 of which determine the sensor that is producing data and the other 3 are sensor data). Each recording is split into overlapping intervals of 32 (with overlap of 16) and all the sequences are combined into one dataset. Out of the total sequences 7769 used for training, 1942 used for testing.

In Table 3, column 2 shows the performance of all the models trained on a person-activity dataset in terms of test accuracy. Our proposed models CDR-NDE and CDR-NDE-heat perform better than all other baseline models. It shows that considering the continuous transformation along both the directions results in a model with better representation capability and generalization performance. We also observe that the more flexible CDR-NDE-heat model using the adaptive Dopri5 solver gives the best performance in the person-activity dataset. To verify the flexibility of the model and the requirement of different depth for different sequences, we computed the number of function evaluations involved while evolving the hidden states over depth in the CDR-NDE-heat(Dopri5) model. We found that the number of function evaluations fall in the range of 26 to 32. This shows that different sequences required different number of function evaluations for learning better representations. Training time for an epoch for the models are CDR-NDE-heat(Euler) : 22 sec, CDR-NDE-Heat(Dopri5) : 30 sec, and CDR-NDE : 58 sec, and shows that CDR-NDE-heat models are faster.

## 5.3 Walker2d kinematic simulation

The dataset was created by (Lechner & Hasani, 2020) for Walker kinematic modeling task. This is a supervised autoregressive task and the dataset was generated using Walker2d-v2 OpenAI gym environment and MuJoCo physics engine. This dataset evaluates how well a model can simulate kinematic modeling systems that are sampled at irregular time intervals. The training data was generated by performing rollouts on the Walker2d-v2 environment using pre-trained deterministic policy. The Walker environment was trained using a non-recurrent policy though Proximal policy optimization before data collection. The dataset is made irregularly sampled by excluding 10% of the timesteps. The dataset is split into 9684 train, 1937 test, 1272 validation sequences.

In Table 3, column 3 shows the performance of all the models on Walker2d data. Our proposed model CDR-NDE-heat(Euler and Dopri5) outperform other models with a good margin. The proposed model CDR-NDE also gives a very good performance in this data. Smoothing of the hidden representations allowed the CDR-NDE-heat model to learn well on this data. Again, the more flexible CDR-NDE-heat model using the adaptive Dopri5 solver gives the best performance. Training time for an epoch for the proposed models are CDR-NDE-heat(Euler) : 28 sec, CDR-NDE-Heat(Dopri5) : 48 sec, and CDR-NDE : 140 sec, and shows that CDR-NDE-heat models are faster.

## 6 Conclusion and Future Work

We proposed novel continuous depth RNN models based on the framework of differential equations. The proposed models generalize recurrent NODE models by continuously evolving in both the temporal and depth directions. CDR-NDE models the evolution of hidden states using two separate differential equations. The CDR-ODE-heat model is designed based on the framework of 1D-Heat equation and models the evolution of the hidden states across time and depth. The experimental results on person activity recognition and Walker2d kinematics data showed that our proposed models outperformed the baselines and are very effective on irregularly sampled real-world sequence data. Currently, CDR-NDE models are designed based on the GRU-cell transformations. We would like to extend it to other transformations as a future work. The continuous depth recurrent neural differential equations are very flexible and generic RNN models. They will have widespread application on several complex sequence modeling and time series problems, involving sequences and inputs with irregular observation times, and varying complexities and modalities.

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
