# OpenReview forum: "Continuous Depth Recurrent Neural Differential Equations"
_ICLR.cc/2023/Conference — Submitted to ICLR 2023_

### Official Review · Reviewer_FygP · 2022-10-20

**Confidence:** 3
**Correctness:** 3
**Technical Novelty And Significance:** 2
**Empirical Novelty And Significance:** 2
**Recommendation:** 3

**Clarity, Quality, Novelty And Reproducibility:**

Overall, the quality of presentation is okay. In some parts it is difficult to follow the arguments (a better organization could help). The ideas presented in this paper are novel, but I don’t feel that the work proposes innovations that are very significant. My main concern is that it is difficult to understand the effects of the different proposed components on the performance, due to a limited discussion and a missing ablation study. Further, I feel that two experiments are not sufficient to demonstrate that this model is useful more generally. The authors do not provide code, to reproduce results. I feel that it would take quite some time to reproduce the results based on just the technical description.

**Strength And Weaknesses:**

The ideas presented in this paper are interesting, and the results demonstrate the effectiveness of the proposed models on 2 tasks. However, I feel that the innovations are somewhat incremental. Moreover, I have several concerns that I would like to see addressed:

* It is not clear to me how the individual components that have been proposed affect the performance, since a detailed ablation study is missing. Is the performance boost due to the specific form of the gates, or due to increasing depth, or because the model has at least twice as many weights as a simpler ODE-GRU. At a minimum it would be helpful to list the parameters for all the models. Then, it would be nice to see how the performance of the proposed model changes as a function of depth.

* You claim that your model is continuously evolving in both the temporal and depth directions, but I am not sure what this means. Where do you show that the model is learning something continuously?  Once you discretize your model for training, you yield a discrete computational graph. There have been several recent works that show that Neural ODEs discretized by explicit Euler do not provide a useful representation for the underlying continuous dynamics of the system of interest. Also, based on your results, it seems that it doesn't matter too much which discretization scheme is used. So, I assume something else must be going on.

* A general problem of RNNs (whether formulated as a discrete or continuous model) are vanishing and exploding gradients during training. Gates certainly mitigate this issue, but it would be good to understand how the proposed model performs on tasks that involve longer sequences.

* For an experimental paper, two sets of experiments that demonstrate the performance of a new model are somewhat weak.

* The paper is missing to discuss several recent state-of-the-art RNNs, based on ODEs, that have appeared in ICML, ICLR, such as coRNN, UnICORNN, incremental RNN,  LEM, and several other related works.

**Summary Of The Paper:**

This paper proposes a continuous formulation of an RNN for modeling sequential data. The contribution of this work is a mechanism to continuously increase depth of recently proposed ODE-based RNN models, as well as a model that is motivated by a 1D-Heat equation. Two benchmark problems (person activity recognition and Walker2d kinematics) are used to show that the proposed models can outperform related models on irregularly sampled sequential data.

**Summary Of The Review:**

In summary, this is an interesting paper with some potential, but I feel that this paper is not ready for publication in its current form. It requires a major revision to better present the impact and significance of the different aspects of the proposed innovations.

---

> ### Author Response · Authors · 2022-12-13
> **Response to the Reviewer FygP**
>
> **Reasons for performance boost**
>
>  From Table 1 and Table 2, it can be observed that the performance of the proposed model CDR-NDE-heat (Dopri5) is better than our another model CDR-NDE-heat (Euler). Adaptive depth definitely played a crucial role in the better performance of  our CDR-NDE-heat (Dopri5) model.
> Our model does not have twice the number of parameters as ODE-GRU, as shown in equation 9 and 11, the same model is evolved in both the temporal and depth directions. We also provide the number of parameters required for each of the baseline and our model. As shown below for most of the baselines the number of learnable parameters are greater than our model, which strengthens the capability of our approach that evolving continuously over time and depth indeed improves the performance of our model.
>
> | Model  | Parameters    |
> | -------|------------|
> |CT-RNN|24K|
> |ODE-RNN|24K|
> |ODE-LSTM|105K|
> |CT-GRU|412K|
> |RNN-Decay|28K|
> |Bidirectional-RNN|142K|
> |GRU-D|73K|
> |Phased-LSTM|97K|
> |GRU-ODE|43K|
> |CT-LSTM|170K|
> |LSTM|97K|
> |Ours|45K|
>
> **Continuous transformation of the hidden representations**
>
> The hidden representations are evolved continuously in both temporal and depth directions, we are approximating the solutions numerically using Euler method in temporal direction and Dopri5 method in depth direction.
> Choice of numerical methods definitely impacts the performance of the model. From Table 1 and Table 2, performance of CDR-NDE-heat (Dopri5) is better than CDR-NDE-heat (Euler), so clearly the discretization scheme does matter.
>
> **Vanishing and Exploding gradients while training with longer sequences**
>
> GRU cell is the building block of our proposed model, our proposed model shares the same benefits and disadvantages of GRU cell. To evaluate the vanishing or exploding gradient problem we will evaluate our model longer sequences.
>
> **More comparisons**
> Thanks for the suggestion, we will make a comparison with the recent related works and include additional datasets.

---

> > ### Author Response · Authors · 2022-12-13
> > **Response to the Reviewer FygP**
> >
> >
> >
> > **Additional Experiements**
> >
> > we  evaluated the performance of the models to predict the stance of social media posts[1] under two different experiment setups. This Twitter data set consists of rumours associated with eight events. Each event has collection of tweets labelled with one of the four labels - Support, Query, Deny and Comment. We picked  Sydneysiege topic to evaluate the models. Given a topic and the corresponding tweets of the sequence, for preparing training data 60\% percentage of the tweets are selected randomly by maintaining the order of the arrival of the tweets, then data points each of length(tweets) 10 are created.  Similarly, the data points for both validation and test data are created by splitting the remaining 40\% equally.  Following are the two experimental setups under which we evaluate the model performances.
> >
> > **Seen Event** Here we train, validate and test on tweets of same event. Each
> > event data is split 60:20:20 ratio in sequence of time. This setup helps in
> > predicting stance of unseen tweets of the same event.
> >
> > **Unseen Event** This setup helps in evaluating performance on an unseen event. Here we consider training and validation on other event(charliehebdo) and testing on sydneysiege.
> >
> > For the seen event experimental setup our proposed models,  CDR-NDE-heat(Euler,Dopri5) as shown in the table below, perform well compared to the baselines in terms of the metrics F1,Recall and Precision.
> >
> > |Model| AUC | F1 | Recall | Precision |
> > |-------------|------------|-------|----------|-----------|
> > |CT-RNN | 0.57 $\pm$ 0.00 | 0.57 $\pm$ 0.00 | 0.70 $\pm$ 0.00 | 0.61 $\pm$ 0.00 |
> > |ODE-RNN | 0.55 $\pm$ 0.01 | 0.55 $\pm$ 0.01 | 0.67 $\pm$ 0.01 | 0.57 $\pm$ 0.01 |
> > |ODE-LSTM | 0.56 $\pm$ 0.01 | 0.56 $\pm$ 0.01 | 0.71 $\pm$ 0.00 | 0.64 $\pm$ 0.01 |
> > |CT-GRU | 0.64 $\pm$ 0.01 | 0.64 $\pm$ 0.01 | 0.72 $\pm$ 0.02 | 0.65 $\pm$ 0.02 |
> > |RNN-Decay | 0.63 $\pm$ 0.01 | 0.63 $\pm$ 0.01 | 0.74 $\pm$ 0.00 | 0.66 $\pm$ 0.00 |
> > |Bidirectional-RNN | 0.62 $\pm$ 0.01 | 0.62 $\pm$ 0.01 | 0.73 $\pm$ 0.01 | 0.66 $\pm$ 0.02 |
> > |GRU-D | 0.64 $\pm$ 0.01 | 0.64 $\pm$ 0.01 | 0.73 $\pm$ 0.01 | 0.65 $\pm$ 0.01 |
> > |Phased-LSTM | 0.61 $\pm$ 0.01 | 0.61 $\pm$ 0.01 | 0.72 $\pm$ 0.01 | 0.62 $\pm$ 0.01 |
> > |GRU-ODE | 0.56 $\pm$ 0.00 | 0.56 $\pm$ 0.00 | 0.70 $\pm$ 0.00 | 0.60 $\pm$ 0.01 |
> > |CT-LSTM | 0.64 $\pm$ 0.01 | 0.64 $\pm$ 0.01 | 0.72 $\pm$ 0.01 | 0.66 $\pm$ 0.01 |
> > |Augmented-LSTM | 0.64 $\pm$ 0.01 | 0.64 $\pm$ 0.01 | 0.73 $\pm$ 0.01 | 0.65 $\pm$ 0.01 |
> > |CDR-NDE | 0.57 $\pm$ 0.01 | 0.62 $\pm$ 0.01 | 0.69 $\pm$ 0.02 | 0.64 $\pm$ 0.05 |
> > |CDR-NDE-heat(Euler) | 0.64 $\pm$ 0.01 | 0.68 $\pm$ 0.01 | 0.73 $\pm$ 0.02 | 0.69 $\pm$ 0.02 |
> > |CDR-NDE-heat(Dopri5) | 0.63 $\pm$ 0.01 | 0.68 $\pm$ 0.01 | 0.73 $\pm$ 0.02 | 0.68 $\pm$ 0.02 |
> >
> > For the unseen event experimental setup our proposed models perform well compared to the baselines in terms of the metrics F1 and Precision, and are comparable under the metrics AUC and Recall.
> >
> > |Model |AUC | F1 | Recall | Precision |
> > |-------------|------------|-------|----------|-----------|
> > |CT-RNN | 0.56 $\pm$ 0.00 | 0.56 $\pm$ 0.00 | 0.70 $\pm$ 0.00 | 0.61 $\pm$ 0.00 |
> > |ODE-RNN | 0.55 $\pm$ 0.00 | 0.55 $\pm$ 0.00 | 0.67 $\pm$ 0.01 | 0.57 $\pm$ 0.01 |
> > |ODE-LSTM | 0.56 $\pm$ 0.00 | 0.56 $\pm$ 0.00 | 0.70 $\pm$ 0.00 | 0.60 $\pm$ 0.01 |
> > |CT-GRU | 0.63 $\pm$ 0.01 | 0.63 $\pm$ 0.01 | 0.75 $\pm$ 0.01 | 0.65 $\pm$ 0.01 |
> > |RNN-Decay | 0.62 $\pm$ 0.00 | 0.62 $\pm$ 0.00 | 0.74 $\pm$ 0.00 | 0.65 $\pm$ 0.01 |
> > |Bidirectional-RNN | 0.61 $\pm$ 0.00 | 0.61 $\pm$ 0.00 | 0.74 $\pm$ 0.00 | 0.67 $\pm$ 0.02 |
> > |GRU-D | 0.63 $\pm$ 0.00 | 0.63 $\pm$ 0.00 | 0.75 $\pm$ 0.00 | 0.67 $\pm$ 0.03 |
> > |Phased-LSTM | 0.58 $\pm$ 0.01 | 0.58 $\pm$ 0.01 | 0.71 $\pm$ 0.01 | 0.65 $\pm$ 0.05 |
> > |GRU-ODE | 0.56 $\pm$ 0.00 | 0.56 $\pm$ 0.00 | 0.70 $\pm$ 0.00 | 0.60 $\pm$ 0.00 |
> > |CT-LSTM | 0.62 $\pm$ 0.00 | 0.62 $\pm$ 0.01 | 0.75 $\pm$ 0.00 | 0.67 $\pm$ 0.01 |
> > |Augmented-LSTM | 0.63 $\pm$ 0.01 | 0.63 $\pm$ 0.01 | 0.75 $\pm$ 0.00 | 0.69 $\pm$ 0.04 |
> > |CDR-NDE  | 0.57 $\pm$ 0.02 | 0.62 $\pm$ 0.04 | 0.67 $\pm$ 0.06 | 0.66 $\pm$ 0.05 |
> > |CDR-NDE-heat(Euler) | 0.62 $\pm$ 0.01 | 0.68 $\pm$ 0.01 | 0.74 $\pm$ 0.01 | 0.69 $\pm$ 0.02 |
> > |CDR-NDE-heat(Dopri5) | 0.62 $\pm$ 0.01 | 0.68 $\pm$ 0.01 | 0.74 $\pm$ 0.01 | 0.71 $\pm$ 0.02 | |
> >
> >
> > [1] Leon Derczynski, Genevieve Gorrell, Arkaitz Zubiaga, Ahmet Aker, Kalina Bontcheva, Maria
> > Liakata, and Elena Kochkina. Rumoureval 2019 data, 2019. URL https://figshare.com/
> > articles/RumourEval_2019_data/8845580/1

---

### Official Review · Reviewer_itQc · 2022-10-21

**Confidence:** 5
**Correctness:** 3
**Technical Novelty And Significance:** 2
**Empirical Novelty And Significance:** 2
**Recommendation:** 3

**Clarity, Quality, Novelty And Reproducibility:**

Regarding novelty, the combination of ODE in time and depth is not a too novel contribution. The main novelties lie in the heat PDE model I belive.

The writing could be sharpened. Specifically, the use of the word "labeling" to describe the processing of data by a neural network is inconsistent with the literature on ML. The common understanding in ML is that labeling describes the process of obtaining targets (i.e., "labels") for the training of the network. Moreover, the use of "and sequence data" in the very first sentence is unnecessary here as "in sequence processing tasks" already require sequence data. More minor things like that make reading the paper unpleasant.


**Strength And Weaknesses:**

# Strenghts
- The paper presents an interesting idea on a relevant topic (time-continuous sequential data).
- The paper presents a novel PDE model for resolving the problem of having two different time variables.

# Weaknesses
The experimental evaluation of the paper is insufficient for ICLR.
First of all, the paper lacks discussion and comparison with significant related works.
For instance, [1,2,3,4,5,6,7] propose models for processing irregularly sampled time series. All of these models have demonstrated significant improvements in continuous-time modeling in the past two years alone.

Moreover, the selection of benchmark datasets is insufficient. Particularly, the paper only evaluates the model on two datasets. Additionally, both evaluations seem to be sourced from (Lechner & Hasani, 2020), which did not perform any hyperparameter tuning, evaluated on five datasets, and does not include any method published after 2020.

[1] Romero et al. CKConv: Continuous Kernel Convolution For Sequential Data. ICLR 2022.
[2] Gu et al. Efficiently Modeling Long Sequences with Structured State Spaces. ICLR 2022.
[3] Kidger et al. Neural Controlled Differential Equations for Irregular Time Series. NeurIPS 2020.
[4] Morrill et al. Neural Rough Differential Equations for Long Time Series. ICML 2020.
[5] Gu et al. HiPPO: Recurrent Memory with Optimal Polynomial Projections. NeurIPS 2020.
[6] Kidger et al. Efficient and Accurate Gradients for Neural SDEs. NeurIPS 2021.
[7] Shukla et al. Multi-Time Attention Networks for Irregularly Sampled Time Series. ICLR 2021.


**Summary Of The Paper:**

The paper proposes a new type of recurrent neural network that operates over continuous time and continuous depth.
The continuous depth is achieved through an auxiliary variable t' that is integrated in each step from zero to a certain Tmax. The paper also proposes a heat equation PDE model to decode both time variables.

**Summary Of The Review:**

Overall, minor but interesting contribution with a bit of unconventional writing and insufficient experimental evaluation.

---

> ### Author Response · Authors · 2022-12-13
> **Response to the Reviewer itQc**
>
> **Comparing with recent works and hyperparameter tuning of baselines**
>
> Thanks for the suggestion, we will make a comparison with the suggested related works. We used the default hyperparameters provided in their code. We will include methods which are recent.

---

### Official Review · Reviewer_Ngo4 · 2022-10-25

**Confidence:** 4
**Correctness:** 2
**Technical Novelty And Significance:** 2
**Empirical Novelty And Significance:** Not applicable
**Recommendation:** 3

**Clarity, Quality, Novelty And Reproducibility:**

**Reproducibility**: Not reproducible. Although the paper includes the hyperparameters, the code to reproduce the results is missing.

**Novelty:** The proposed ideas of learning continuous depth model as well as using second order partial differential equation to evolve hidden states are novel and interesting. It is not well motivated though.

**Strength And Weaknesses:**

### Strengths
1. The proposed ideas of learning continuous depth model as well as using second order partial differential equation to evolve hidden states are novel and interesting.
2.  Experimental results show the effectiveness of the approach when compared to baseline approaches.
3. The paper focuses on the task of learning from irregularly sampled data which is important in many domains.

### Weaknesses
1. Although it is interesting work, it is not well motivated. Why would continuous in depth potentially be useful for modeling irregularly sampled time series data?
2. Another key concern about the paper is the lack of rigorous experimentation to study the usefulness of the proposed method. The paper only experiments on a single real-world dataset and missing experimentation of benchmark datasets (e.g. PhysioNet, MIMIC-III).
3. The paper is missing important related work and comparison with the SOTA in modeling irregularly sampled time series [1]. The paper also seems to underperform compared to [1] on person activity recognition.
4. In the human activity dataset, which is a per time-step classification problem, it seems that the sequence was fed into the proposed model as a whole, thus the model has access to future observations for making a classification prediction on "past" time steps. This would mean that it has an unfair advantage over the RNN based models.
5. RNN and ODE based recurrent approaches can be used for whole-time series classification, imputation, interpolation and extrapolation tasks. The paper only focuses on per time point classification/regression task which is very limiting. It is also not clear how the run time of the proposed approach compare to that of ODE-RNN and other baselines.


#### References
1. Satya Narayan Shukla and Benjamin Marlin. Multi-time attention networks for irregularly sampled time series. In International Conference on Learning Representations, 2021.


**Summary Of The Paper:**

The paper focuses on learning of multivariate irregularly sampled time series data. Recently, several works have introduced continuous version of RNNs based on Neural ODEs to solve the problem of irregular sampling. This paper generalizes the recurrent Neural ODE models by continuously evolving the hidden states in both temporal and depth directions. It employs two separate differential equations to evolve the hidden states in both horizontal and vertical direction alternatively. The paper also proposes heat equation based model which treats the hidden state computation as solving heat equation over time. This model is also able to learn a better representation by utilizing information from past as well as future. Experiments show that the proposed models outperform the RNNs and neural ODE based models on two datasets.

**Summary Of The Review:**

My recommendation for this paper is reject. Although the paper proposes a novel solution, motivation is lacking behind the proposed approach, rigorous experimentation is missing.

---

> ### Author Response · Authors · 2022-12-13
> **Response to the Reviewer Ngo4**
>
> **Why continuity in depth is needed**
>
> For a given sequence it might require more transformations depending on the complexity of the sequence. Continuous depth allows flexibility in modeling sequence data, with different depths over the elements in the sequence as well as different sequences. Combining this with the continuous time transformation as in recurrent neural ODE allows greater modeling capability for irregularly sampled time series data. To verify that different sequences required different depth, for CDR-NDE-Heat(Dopri5) model on Person-Activity dataset we computed the number of function evaluations involved while evolving the hidden states over depth. We found that the number of function evaluations required fall in the range of 26 to 32. This shows that different sequences required different number of function evaluations for learning better representation.
>
>
>
> **Access to future observations**
>
> This is because of the specific property of the CDR-NDE-heat model. The model is capable of looking ahead due to the nature of the heat equation and consider  latent  representations  from the future (h(t+∆t,t′)). This provides a better modeling capability than standard RNNs
>
> **Possibility of interpolation, extrapolation, imputation tasks**
>
> We believe the proposed approach is going to be most useful for per time point classification/regression tasks having sequences and inputs with different complexities. However it is not a limitation of the approach and can be used for whole-time series classification, imputation, interpolation and extrapolation tasks similar to ODE-RNNs.

---

> > ### Author Response · Authors · 2022-12-13
> > **Response to the Reviewer Ngo4**
> >
> > **Additional Experiments**
> >
> > Thank you for the suggestion. We have demonstrated the effectiveness of our models on two data sets, Person Activity Recognition and Walker2d Kinematic. We will consider the suggestion of demonstrating the effectiveness on  PhysioNet and  MIMIC-III.
> >
> > We  evaluated the performance of the models to predict the stance of social media posts[1] under two different experiment setups. This Twitter data set consists of rumours associated with eight events. Each event has collection of tweets labelled with one of the four labels - Support, Query, Deny and Comment. We picked  Sydneysiege topic to evaluate the models. Given a topic and the corresponding tweets of the sequence, for preparing training data 60\% percentage of the tweets are selected randomly by maintaining the order of the arrival of the tweets, then data points each of length(tweets) 10 are created.  Similarly, the data points for both validation and test data are created by splitting the remaining 40\% equally.  Following are the two experimental setups under which we evaluate the model performances.
> >
> > **Seen Event** Here we train, validate and test on tweets of same event. Each
> > event data is split 60:20:20 ratio in sequence of time. This setup helps in
> > predicting stance of unseen tweets of the same event.
> >
> > **Unseen Event** This setup helps in evaluating performance on an unseen event. Here we consider training and validation on other event(charliehebdo) and testing on sydneysiege.
> >
> > For the seen event experimental setup our proposed models,  CDR-NDE-heat(Euler,Dopri5) as shown in the table below, perform well compared to the baselines in terms of the metrics F1,Recall and Precision.
> >
> > |Model| AUC | F1 | Recall | Precision |
> > |-------------|------------|-------|----------|-----------|
> > |CT-RNN | 0.57 $\pm$ 0.00 | 0.57 $\pm$ 0.00 | 0.70 $\pm$ 0.00 | 0.61 $\pm$ 0.00 |
> > |ODE-RNN | 0.55 $\pm$ 0.01 | 0.55 $\pm$ 0.01 | 0.67 $\pm$ 0.01 | 0.57 $\pm$ 0.01 |
> > |ODE-LSTM | 0.56 $\pm$ 0.01 | 0.56 $\pm$ 0.01 | 0.71 $\pm$ 0.00 | 0.64 $\pm$ 0.01 |
> > |CT-GRU | 0.64 $\pm$ 0.01 | 0.64 $\pm$ 0.01 | 0.72 $\pm$ 0.02 | 0.65 $\pm$ 0.02 |
> > |RNN-Decay | 0.63 $\pm$ 0.01 | 0.63 $\pm$ 0.01 | 0.74 $\pm$ 0.00 | 0.66 $\pm$ 0.00 |
> > |Bidirectional-RNN | 0.62 $\pm$ 0.01 | 0.62 $\pm$ 0.01 | 0.73 $\pm$ 0.01 | 0.66 $\pm$ 0.02 |
> > |GRU-D | 0.64 $\pm$ 0.01 | 0.64 $\pm$ 0.01 | 0.73 $\pm$ 0.01 | 0.65 $\pm$ 0.01 |
> > |Phased-LSTM | 0.61 $\pm$ 0.01 | 0.61 $\pm$ 0.01 | 0.72 $\pm$ 0.01 | 0.62 $\pm$ 0.01 |
> > |GRU-ODE | 0.56 $\pm$ 0.00 | 0.56 $\pm$ 0.00 | 0.70 $\pm$ 0.00 | 0.60 $\pm$ 0.01 |
> > |CT-LSTM | 0.64 $\pm$ 0.01 | 0.64 $\pm$ 0.01 | 0.72 $\pm$ 0.01 | 0.66 $\pm$ 0.01 |
> > |Augmented-LSTM | 0.64 $\pm$ 0.01 | 0.64 $\pm$ 0.01 | 0.73 $\pm$ 0.01 | 0.65 $\pm$ 0.01 |
> > |CDR-NDE | 0.57 $\pm$ 0.01 | 0.62 $\pm$ 0.01 | 0.69 $\pm$ 0.02 | 0.64 $\pm$ 0.05 |
> > |CDR-NDE-heat(Euler) | 0.64 $\pm$ 0.01 | 0.68 $\pm$ 0.01 | 0.73 $\pm$ 0.02 | 0.69 $\pm$ 0.02 |
> > |CDR-NDE-heat(Dopri5) | 0.63 $\pm$ 0.01 | 0.68 $\pm$ 0.01 | 0.73 $\pm$ 0.02 | 0.68 $\pm$ 0.02 |
> >
> > For the unseen event experimental setup our proposed models perform well compared to the baselines in terms of the metrics F1 and Precision, and are comparable under the metrics AUC and Recall.
> >
> > |Model |AUC | F1 | Recall | Precision |
> > |-------------|------------|-------|----------|-----------|
> > |CT-RNN | 0.56 $\pm$ 0.00 | 0.56 $\pm$ 0.00 | 0.70 $\pm$ 0.00 | 0.61 $\pm$ 0.00 |
> > |ODE-RNN | 0.55 $\pm$ 0.00 | 0.55 $\pm$ 0.00 | 0.67 $\pm$ 0.01 | 0.57 $\pm$ 0.01 |
> > |ODE-LSTM | 0.56 $\pm$ 0.00 | 0.56 $\pm$ 0.00 | 0.70 $\pm$ 0.00 | 0.60 $\pm$ 0.01 |
> > |CT-GRU | 0.63 $\pm$ 0.01 | 0.63 $\pm$ 0.01 | 0.75 $\pm$ 0.01 | 0.65 $\pm$ 0.01 |
> > |RNN-Decay | 0.62 $\pm$ 0.00 | 0.62 $\pm$ 0.00 | 0.74 $\pm$ 0.00 | 0.65 $\pm$ 0.01 |
> > |Bidirectional-RNN | 0.61 $\pm$ 0.00 | 0.61 $\pm$ 0.00 | 0.74 $\pm$ 0.00 | 0.67 $\pm$ 0.02 |
> > |GRU-D | 0.63 $\pm$ 0.00 | 0.63 $\pm$ 0.00 | 0.75 $\pm$ 0.00 | 0.67 $\pm$ 0.03 |
> > |Phased-LSTM | 0.58 $\pm$ 0.01 | 0.58 $\pm$ 0.01 | 0.71 $\pm$ 0.01 | 0.65 $\pm$ 0.05 |
> > |GRU-ODE | 0.56 $\pm$ 0.00 | 0.56 $\pm$ 0.00 | 0.70 $\pm$ 0.00 | 0.60 $\pm$ 0.00 |
> > |CT-LSTM | 0.62 $\pm$ 0.00 | 0.62 $\pm$ 0.01 | 0.75 $\pm$ 0.00 | 0.67 $\pm$ 0.01 |
> > |Augmented-LSTM | 0.63 $\pm$ 0.01 | 0.63 $\pm$ 0.01 | 0.75 $\pm$ 0.00 | 0.69 $\pm$ 0.04 |
> > |CDR-NDE  | 0.57 $\pm$ 0.02 | 0.62 $\pm$ 0.04 | 0.67 $\pm$ 0.06 | 0.66 $\pm$ 0.05 |
> > |CDR-NDE-heat(Euler) | 0.62 $\pm$ 0.01 | 0.68 $\pm$ 0.01 | 0.74 $\pm$ 0.01 | 0.69 $\pm$ 0.02 |
> > |CDR-NDE-heat(Dopri5) | 0.62 $\pm$ 0.01 | 0.68 $\pm$ 0.01 | 0.74 $\pm$ 0.01 | 0.71 $\pm$ 0.02 | |
> >
> >
> > [1] Leon Derczynski, Genevieve Gorrell, Arkaitz Zubiaga, Ahmet Aker, Kalina Bontcheva, Maria
> > Liakata, and Elena Kochkina. Rumoureval 2019 data, 2019. URL https://figshare.com/
> > articles/RumourEval_2019_data/8845580/1

---

### Decision · Program_Chairs · 2023-01-20

**Decision:**

Reject

**Justification For Why Not Higher Score:**

N/A

**Justification For Why Not Lower Score:**

N/A

**Metareview: Summary, Strengths And Weaknesses:**

This paper presents a novel RNN that operates in both continuous time and continuous depth. An auxiliary variable, t', is introduced and integrated from 0 to a maximum Tmax, resulting in a continuous depth. Furthermore, a heat equation PDE model is proposed to decode the time variables.

Strength: This paper introduces an interesting concept related to time-continuous sequential data, with a proposed PDE model to tackle the issue of dual temporal variables.

The weaknesses of the paper include the lack of reproducibility due to the absence of code, lack of motivation for the proposed approach, insufficient discussion and comparison with significant related works, and insufficient selection of benchmark datasets.

**Summary Of Ac-Reviewer Meeting:**

N/A